# Getting Better or Getting Worse? A Population-Based Study on Trends in Self-Rated Health among Single Mothers in Germany between 1994 and 2018

**DOI:** 10.3390/ijerph19052727

**Published:** 2022-02-26

**Authors:** Stefanie Sperlich, Frauke-Marie Adler, Johannes Beller, Batoul Safieddine, Juliane Tetzlaff, Fabian Tetzlaff, Siegfried Geyer

**Affiliations:** Medical Sociology Unit, Hannover Medical School, 30625 Hannover, Germany; frauke-marie.adler@stud.mh-hannover.de (F.-M.A.); beller.johannes@mh-hannover.de (J.B.); safieddine.batoul@mh-hannover.de (B.S.); tetzlaff.juliane@mh-hannover.de (J.T.); tetzlaff.fabian@mh-hannover.de (F.T.); geyer.siegfried@mh-hannover.de (S.G.)

**Keywords:** self-rated health, health inequalities, public health, trend, single motherhood, single parenthood

## Abstract

Background: While numerous studies suggest that single motherhood is associated with socioeconomic disadvantages and poor health, few studies have analyzed how these conditions have evolved over time. Addressing this gap, we examined the temporal development of self-rated health (SRH) among single compared to partnered mothers, and the role of socioeconomic factors that may have influenced this trend. Methods: We used representative longitudinal data from the German Socioeconomic Panel Survey (G-SOEP) between 1994 and 2018, consisting of 83,843 women with children, aged 30–49 years (13,664 single and 70,179 partnered mothers). Time trends in SRH and socioeconomic factors were analyzed by means of logistic regression analyses. We applied the Karlson–Holm–Breen (KHB) method for decomposing the total time effect into direct and indirect parts via socioeconomic mediators. Results: The predicted probabilities of good SRH decreased in single mothers from 57.0% to 48.4%, while they increased in partnered mothers from 54.8% to 61.3%. Similarly, predicted probabilities of poor SRH rose from 15.0% to 22.7% in single mothers while decreasing slightly from 12.0% to 11.4% in partnered mothers. Moreover, socioeconomic factors worsened over time for single mothers, while they mostly improved for partnered mothers. Decomposing the time trend revealed that the deterioration of single mothers’ health was partly explained by the worsening of socioeconomic disadvantages, of which the decline in full-time employment, the rise in low incomes, and in unemployment contributed most. Conclusions: The alarming rise in socioeconomic and health disadvantages among single mothers in Germany shows that action is needed to counter this trend.

## 1. Introduction

As in other Western countries [1,2], the number of single-parent families is on the rise in Germany. In 2020, the proportion of single parenthood in all families was 17.3%, while two decades ago the share was 15.3%. Single parenthood is strongly gendered, and the high proportion of single mothers in Germany has decreased only slightly from 88% to 85% over the last 20 years [3]. Numerous studies have demonstrated that single motherhood is associated with adverse health outcomes, such as depressive symptoms and anxiety [4,5,6,7,8], disabilities [9], and poor self-rated health [6,10,11,12]. Berkman et al. [9] pointed out that higher health risks among single mothers are not limited to young and middle adulthood but are also present at older ages. Consistent with this, studies have shown that single parents also have increased risks for severe morbidity and mortality [13,14,15]. The elevated health risks of single mothers are largely explained by psychosocial distress due to sole parenting, difficulties of reconciling work and family life, and the lack of social support and socioeconomic strains such as unemployment and financial hardship [16,17].

While many studies provide evidence that single motherhood is associated with socioeconomic disadvantages and poorer health, few studies have been conducted on how these conditions have evolved over time. Burström et al. [18] found that the socioeconomic conditions of single mothers in Sweden deteriorated over the years 1979–1995, as employment rates declined and poverty rates increased. Fritzell and Burström [19] analyzed the implications of changes on the labor market and in the Swedish welfare policy during the 1990s for single mothers. They found that prevalence rates of economic strain and poor SRH increased during the 1990s compared to the 1980s among both single and partnered mothers. The difference in poor SRH between motherhood types increased during the later time period as SRH of single mothers deteriorated at a higher pace. In addition, Fritzell et al. [20] found that employment rates among Swedish single mothers declined from 1983 to 2001, while the prevalence of self-reported financial problems increased. However, they found no evidence of increased health differentials between single and partnered mothers in terms of poor SRH, hospitalization or mortality. Trujillo-Alemán et al. [21] investigated whether health inequalities between single and partnered mothers changed in Spain before and during the financial crisis. Comparing the periods 2003–2004 and 2011–2012, they found some evidence of increasing inequalities between single and partnered mothers for the manual social class with respect to chronic conditions.

We are not aware of any study from Germany that has analyzed the direction in which the health of single mothers has developed over the last decades. Turning to this research question is particularly interesting in the light of the many preventive efforts that have been targeted at single parents in recent years [6]. The present study addresses this issue and examines the development of SRH among single compared to partnered mothers in Germany between 1994 and 2018. In more detail, the study is guided by the following research questions:

How has SRH developed in single compared to partnered mothers?How have socioeconomic living conditions in terms of school education, employment status, occupational position and income changed for single mothers relative to partnered mothers?Can the temporal development of SRH in single mothers be explained by changes in socioeconomic living conditions?

## 2. Materials and Methods

### 2.1. Data

We used longitudinal data from the German Socioeconomic Panel (GSOEP V.31). The GSOEP is a representative annual survey of the German population based on face-to-face interviews in private households, conducted by the German Institute for Economic Research (DIW Berlin) and the Kantar Group [22]. Some of the many topics surveyed include household composition, employment status, income, occupation and health. The survey uses random probability samples based on a nation-wide two-stage stratified sampling procedure. Further information on GSOEP can be obtained from Goebel et al. [22]. 

Our analyses were based on a pooled dataset including the waves from 1994 to 2018, allowing for trend analysis on a population level by means of cross-sectional comparison. In principle, single-parenthood can extend from teenage years to retirement age. However, previous studies have shown that trends in health differed markedly depending on the age group and life stage considered [23,24]. Therefore, we focused on the age group 30–49 years, as it is mainly concerned with family obligations and included 77.1% of all single mothers in the sample. We used cross-sectional weights that were assumed to produce a nationally representative sample [25]. Respondents with missing information were excluded from the analysis.

### 2.2. Measures

The participants were asked to rate their general health status on a five-response scale (‘very good’, ‘good’, ‘satisfactory’, ‘poor’, and ‘bad’). We transformed this variable into two binary variables indicating ‘good’ (very good/good vs. the other categories) and ‘poor’ health (poor/bad vs. the other categories). SRH has proven to be a reliable and valid health indicator that is associated with healthcare utilization, future health problems, multiple biomarkers and mortality [26,27,28]. The time trend as the independent variable was assessed by means of a categorical variable covering five time periods (1994–1998, 1999–2003, 2004–2008, 2009–2013 and 2014–2018), using the first time period as a reference category. In addition, we used a continuous trend variable, coded 0 for 1994 and 1 for 2018, with the years in between getting fractional values.

We defined single mothers as women with at least one underage child living in the household without a partner. Accordingly, we defined partnered mothers as females cohabitating with a partner with at least one underage child, regardless of being married or not.

Socioeconomic factors (school education, occupational position and household income) were classified into three categories, representing low, intermediate and high social status (see Table 1). Employment status was categorized into ‘unemployed and seeking for work’, ‘not employed and not seeking for work (e.g., parental leave)’ ‘employed part-time’ and ‘employed full-time’.

### 2.3. Statistical Analyses

We tested the effect of time on SRH among single and partnered mothers by means of logistic regression analyses using cluster-robust standard errors to adjust for the panel structure of the data. Interaction terms were calculated between the trend variable and family status (single versus partnered mothers) in order to determine changes in health inequality among single mothers, using ‘first year of observation’ and ‘partnered mothers’ as the reference categories. The Karlson–Holm–Breen method [29] (KHB) was applied to estimate the effects of socioeconomic factors as possible mediators of the time-effect on single mothers’ SRH. The KHB method extends the decomposition properties of linear models to logistic regression models by decomposing the total effect of time on SRH into a direct and indirect component. This method ensures that the crude and adjusted coefficients presented are measured on the same scale and, hence, are unaffected by the rescaling biases that arise in cross-model comparisons of non-linear models. The rescaling bias means that the inclusion of the mediator variable Z in a nonlinear probability model will change the coefficient of X regardless of whether Z is correlated with X, it is a sufficient condition that Z is correlated with Y. The idea of the KHB method is to extract from Z the information that is not contained in X by calculating the residuals of a linear regression of Z on X and to use them for calculating the total effect [29].

In our case, the total time effect is the effect of time on SRH only controlled for age and the residuals. The direct time effect corresponds to the effect that remains after additionally controlling for socioeconomic factors as potential mediators. Accordingly, the indirect effect is the part of the time effect on SRH that is explained by socioeconomic factors. In case of odds ratios (OR), the indirect effect is calculated as the total effect divided by the direct effect. In addition to OR, we reported average partial effects (APEs) giving the decomposition a more substantial interpretation. APEs are measured on the probability scale and estimate the average marginal effect of each mediator [29]. In the present case, APEs provide information on the change in the average probability of good/poor SRH between 1994 and 2018, expressed in percentage points. The indirect time effect is calculated as the difference between the total time effect and the direct time effect. A description of this method and the user-written program ‘khb’ can be found in Kohler et al. [29].

We used the STROBE cross sectional reporting guidelines [30]. All analyses were performed with STATA v13.1.

## 3. Results

### 3.1. Sociodemographic Characteristics in Single and Partnered Mothers

In total, 3758 single and 14,338 partnered mothers were surveyed 13,664 and 70,179 times between 1994 and 2018, respectively. The weighted sample characteristics, separated by gender and time period, are presented in Table 2. As displayed, the proportion of missing data was low, with the exception of the income variable, with 8.9% missing values for single mothers.

Single mothers compared to partnered mothers were disadvantaged in terms of school education, employment status, occupational position and household income (Table 1). Between 1994 and 2018, predicted probabilities of being a single mother increased from 8.2% to 9.8%, while probabilities for being a partnered mother decreased from 54.0% to 49.0%. Accordingly, the share of childless women increased from 37.8% to 41.2% over time. Exclusively considering women with underage children, the predicted probabilities of being a single mother increased from 13.0% to 16.0% (Table 3).

### 3.2. Time Trends in SRH in Single and Partnered Mothers

Over time, the predicted probabilities of good SRH decreased in single mothers from 57.0% to 48.4%, while they increased in partnered mothers from 54.8% to 61.3% (Figure 1). Expressed in odds ratios (OR), the chance of good SRH decreased in single mothers over time by 32% (OR: 0.68, CI: 0.50–0.93), while it increased in partnered mothers by 39% (OR: 1.39, CI: 1.22–1.59) (Table 4). In addition, odds of poor SRH increased among single mothers (OR: 1.86, CI: 1.26–2.76) while they remained largely unchanged in partnered mothers (OR: 0.96, CI: 0.76–1.16). As indicated by the interaction terms, the diverging time trends in SRH between single and partnered mothers were statistically significant (Table 4). 

### 3.3. Trends in Socioeconomic Factors in Single and Partnered Mothers

The chance of attaining a high educational level has increased significantly among partnered mothers but not among single mothers (Table 5). The proportion of unemployment increased for single mothers but decreased in partnered mothers. Moreover, the proportion of a high occupational position did not change significantly for single mothers, while it rose in partnered mothers. In terms of employment status, it appeared that the share of part-time employment increased while the proportion of-full time employment decreased for both single and partnered mothers. The share of a low income also increased for both single and partnered mothers. However, as Table 6 illustrates, for single mothers, who started at a substantial higher level, the share increased to 41.6% as compared to 10.3% in partnered mothers. 

### 3.4. Decomposition of the Time Effect on SRH in Single Mothers

Between 1994 and 2018, the chance of good SRH decreased in single mothers (Table 7, total time effect). Expressed in APE, the probability of good SRH decreased by 12.0% points (total time effect). After controlling for socioeconomic factors, the decrease was reduced to 8.6% points (direct time effect). The mediators thus contributed to a decrease of 3.4% points (indirect time effect). This corresponds to a share of 28.2% of the total decrease attributable to socioeconomic factors.

The same picture emerged with respect to poor SRH where average probabilities increased in single mothers by 9.9% points. After controlling for socioeconomic factors, the increase reduced to 7.3% points. Accordingly, the mediators contributed by 2.6% points to this increase, corresponding to a share of 26.4% of the total rise in poor SRH.

Disentangling the contributions of each socioeconomic factor revealed that the decrease in full-time employment with a share of 32.1% of the indirect effect contributed most to the deterioration of good SRH in single mothers (Table 7, lower part). In addition, the rise in low income (25.9%) and in unemployment (24.8%), as well as the decreasing share of high occupational positions (23.2%) contributed to single mothers’ decline in good health over time. The only exception that pointed towards improvements over time was the decline in low education leading to an increase in the average probabilities of good SRH by 0.6% points. A very similar picture emerged with regard to poor SRH.

## 4. Discussion

The key finding of our study is that health inequalities between single and partnered mothers are on the rise to the disadvantage of single mothers. Furthermore, we found that approximately a quarter of the deterioration in single mothers’ SRH could be explained by the expansion of social disadvantages over time.

### 4.1. Trends in Single Mothers’ SRH and the Contribution of Socioeconomic Factors

Numerous studies have demonstrated that single motherhood is associated with poorer health [4,5,6,9,10,11,12,31]. However, only few studies provide information on the development of health disparities among single and partnered mothers over time. While the study by Fritzell et al. [20] revealed that social and health disparities between single and partnered mothers remained largely unchanged, other studies pointed to increasing disadvantages in single mothers [19,21]. In line with the latter findings, we found widening health inequalities between single and partnered mothers. Moreover, we also found that socioeconomic disadvantages were on the rise and that approximately 25–30% of the decline in SRH could be attributed to these increases in social disadvantages. The decline in full-time employment contributed most to the deterioration in SRH. In addition, the rise in low incomes and in unemployment rates as well as the decreasing shares of high occupational positions were also important drivers of this decline. However, a substantial part of the time effect remains that could not be explained by socioeconomic factors. This finding is consistent with other studies revealing that health disparities in single mothers did not completely disappear after controlling for socioeconomic factors [6,20,32]. Previous studies have demonstrated that, in addition to socioeconomic status, the availability of social support, the compatibility of work and family demands, and the relationship with the children as well with the ex-partner are of particular importance for single mothers’ health [16,33,34,35]. Moreover, research has shown that single mothers report more sadness and stress in parenting [36], as well as lower levels of health literacy [37]. The temporal development of these factors and their influence on single mothers’ health should therefore be investigated in further studies.

### 4.2. Trends in Single Mothers’ Health in the Light of Preventive Measures and Welfare Reforms

In recent years, single mothers have increasingly gained attention in public health research and there is a rising awareness of their particular risk of poverty and poor health. Subsequently, a number of family-related policies have been employed in order to tackle financial hardships among single parents such as subsidized childcare, tax relief, and children’s and parental allowances [6]. By expanding childcare hours and improving opportunities in the labor market, single parents should be enabled to work full-time and thus improve their financial situation. In order to advance this ‘welfare-to-work policy’ the German maintenance reform from the year 2008 decided that single parents are generally obliged to return to work, even full-time, when the youngest child has reached the age of three. As our study revealed that full-time employment was declining in single mothers, while unemployment and low-incomes were on the rise, the success of these welfare-to-work efforts could be judged critically. Our finding that the decline in full-time employment over time contributed most to single mothers’ health deterioration appears reasonable, as full-time employment is particularly important for single mothers’ financial independency. However, reverse causality may also play a role in the sense that deteriorating health makes it increasingly hard for single mothers to work full-time. Moreover, while full-time employment reduces the risk of poverty, it may exacerbate the problem of reconciling work and family at the same time, as even less time would be available for the household and the family. A systematic review of health consequences of welfare-to-work policies revealed that they can result in increased conflicts and reduced control, which may have negative impacts on mental health [38]. Setting different priorities than in the welfare-to-work policy, Bertram [39] emphasize that family policy measures must be embedded in corresponding concepts of family time budgets. He advocate for a reduction in working hours with financial loss compensation in the life-phase of rearing children in order to be able to meet the family needs. Since our studies indicate that neither the socioeconomic nor the health situation of single parents has improved, such alternative approaches seem worth considering.

## 5. Limitations

Finally, some important limitations of the study need to be addressed. SRH has proven to be a reliable and valid health indicator [26,27,28]. However, caution is advised when comparing health ratings of different subgroups such as single and partnered mothers, as some of the differences may be due to systematic reporting tendencies. Moreover, since previous studies have shown that trends in health differed markedly from the life stage considered [23,24], we focused on the age group 30–49 years, that is mainly concerned with family responsibilities. With this approach, we captured 77%: the majority of single mothers. However, we cannot make any statements about younger or older single mothers, who should be considered in further analyses. In addition, single mothers differ in terms of the father’s involvement in raising the children. The fact of whether the children resided with the mother or were regularly left overnight with the father was not surveyed in the study, and accordingly could not be taken into account. However, it is to be expected that this has an impact on the living situation of single mothers and their burdens. In addition, though sampling weights were used, the existence of sampling bias cannot be completely ruled out. Selection bias might also be due to the exclusion of women who could not participate in the survey for health reasons. Therefore, we cannot exclude that SRH is overestimated in our study. However, there is no reason to assume that the proportion of non-participating women changed substantially over time. Hence, the time trends reported should not be affected by this source of bias.

## 6. Conclusions

Despite numerous policies that have been implemented in recent years to support single parents, we found that single mothers’ SRH has deteriorated over the last decades, leading to widening health inequalities between single and partnered mothers. The deterioration in single mothers’ SRH is partly due to their increasing socioeconomic disadvantages. The alarming decline in single mothers’ SRH suggests that previous policies to support this vulnerable group should be reconsidered.

## Figures and Tables

**Figure 1 ijerph-19-02727-f001:**
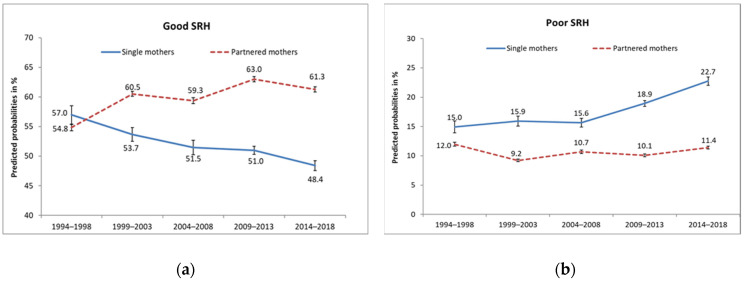
Predicted probabilities and standard errors of (**a**) good and (**b**) poor SRH from 1994–1998 to 2014–2018 in single and partnered mothers, Germany, adjusted for age.

**Table 1 ijerph-19-02727-t001:** Operationalization of the socioeconomic status.

Indicator	Socioeconomic Status		
	Low	Intermediate	High
School education	No school leaving certificate or maximum 9 years of schooling	10 years of schooling	12–13 years of schooling
Occupational position	Unskilled, semi-skilled and skilled workers, farmers, salaried employees with simple tasks and civil servants in the ordinary service	Self-employed persons without employees, salaried employees with qualified tasks and civil servants in the middle civil service	Self-employed persons with employees, salaried employees with highly qualified jobs, master/ mistress, civil servants in the upper and higher levels of the civil service
Household income ^1^	<60% of the median income (poverty risk threshold)	Between 60% and 100% of the median income	>100% of the median income

^1^ Based on modified OECD equivalence scale.

**Table 2 ijerph-19-02727-t002:** Weighted sample characteristics of single and partnered mothers aged 30–49 years in Germany between 1994 and 2018 (in %).

Sample Characteristics	Single Mothers (*n* = 13,664)	Partnered Mothers (*n* = 70,179)
Age groups in yrs.		
30–34	21.2	24.2
35–39	29.8	31.6
40–44	28.2	28.4
45–49	20.8	15.8
Missing	0	0
Number of children		
1	63.4	44.2
2	27.3	42.1
3+	9.2	13.7
Missing	0	0
Age of children ^1^		
0–4	14.7	30.2
5–10	42.8	49.7
11–18	65.2	56.0
Missing	0	0
School education		
Primary	30.8	24.8
Secondary	39.1	38.3
Tertiary	20.5	25.7
Other qualification	8.6	9.8
Missing	1	1.3
Employment status		
Unemployed	15.3	4.5
Not employed	15.4	32.0
Part-time	37.0	44.8
Full-time	32.0	18.8
Missing	0	0
Occupational position ^2^		
Low	18.1	15.3
Intermediate	40.6	37.9
High	10.9	11.3
Not working	30.5	35.6
Missing	0	0
Household income		
<60% median income	32.3	7.1
60%–<150%	56.2	76.5
≥150%	2.5	15.2
Missing	8.9	1.2

Notes: *n* = number of observations. ^1^ having at least one child of that age, ^2^ categories low, intermediate and high occupational position are explained in Table 1.

**Table 3 ijerph-19-02727-t003:** Development of family status in women aged 30–49 years, Germany, 1994–2018.

Time	All Women									Women with Children					
	Single Mothers (*n* = 13,664)			Partnered Mothers (*n* = 70,179)			Childless Women (*n* = 23,116)			Single Mothers (*n* = 13,664)			Partnered Mothers (*n* = 70,179)		
	%	OR	95% CI	%	OR	95% CI	%	OR	95% CI	%	OR	95% CI	%	OR	95% CI
Model 1:															
1994–1998	8.2	1		54.0	1		37.8	1		13.0	1		87.0	1	
1999–2003	9.5	1.18 *	1.05; 1.57	52.9	0.95	0.88; 1.04	37.6	0.99	0.91; 1.08	15.0	1.18 *	1.00; 1.40	85.0	0.84 *	0.72; 0.99
2004–2008	10.3	1.28 *	0.96; 1.57	50.0	0.85 **	0.76; 0.95	39.8	1.09	0.97; 1.23	16.6	1.34 **	1.09; 1.65	83.4	0.75 **	0.61; 0.92
2009–2013	9.3	1.15	0.95; 1.38	46.2	0.72 ***	0.65; 0.81	44.5	1.33 ***	1.18; 1.50	16.5	1.32 **	1.09; 1.61	83.5	0.75 **	0.62; 0.92
2014–2018	9.8	1.21 *	1.00; 1.47	49.0	0.82 **	0.73; 0.92	41.2	1.16 *	1.02; 1.31	16.0	1.28 *	1.05; 1.58	84.0	0.78 *	0.63; 0.95
Model 2:															
1994–2018 (cont.)		1.20	0.97; 1.47		0.72 ***	0.63; 0.82		1.32 ***	1.15; 1.53		1.35 **	1.09; 1.68		0.74 **	0.60; 0.92

Notes: adjusted for age, % = predicted probabilities, OR = odds ratio, 95% CI = 95% confidence interval. Model 1: categorical time variable, Model 2: interval scaled time variable with first year of observation (1994) as reference category, * *p* ≤ 0.05, ** *p* ≤ 0.01, *** *p* ≤ 0.001.

**Table 4 ijerph-19-02727-t004:** Development of good/poor self-rated health (SRH) in single and partnered mothers aged 30–49 years in Germany, 1994–2018.

Time	Single Mothers (*n_max_* = 13,664)		Partnered Mothers (*n_max_* = 70,179)		Interaction Term (*n_max_* = 83,843)	
	OR	95% CI	OR	95% CI	OR	95% CI
Good SRH						
Model 1						
1994–1998	1		1		1	
1999–2003	0.87	0.59; 1.08	1.27 ***	1.16; 1.39	0.71 *	0.53; 0.94
2004–2008	0.80	0.60; 1.08	1.21 **	1.07; 1.35	0.67 *	0.48; 0.92
2009–2013	0.79	0.60; 1.03	1.41 ***	1.25; 1.58	0.59 **	0.44; 0.79
2014–2018	0.71 *	0.53; 0.94	1.31 ***	1.17; 1.47	0.56 **	0.41; 0.76
Model 2						
Time (cont.)	0.68 *	0.50; 0.93	1.39 ***	1.22; 1.59	0.52 ***	0.37; 0.73
Poor SRH						
Model 1						
1994–1998	1		1		1	
1999–2003	1.08	0.75; 1.55	0.75	0.65; 0.86	1.37	0.92; 2.05
2004–2008	1.06	0.73; 1.53	0.88	0.74; 1.94	1.16	0.76; 1.77
2009–2013	1.33	0.95; 1.86	0.82 *	0.70; 0.97	1.53 *	1.04; 2.24
2014–2018	1.68 **	1.18; 2.38	0.95	0.80; 1.11	1.70 **	1.15; 2.52
Model 2						
Time (cont.)	1.86 **	1.26; 2.76	0.96	0.79; 1.16	1.87 **	1.20; 2.92

Notes: Based on logistic regression analyses adjusted for age, good SRH = response category ‘very good’ and ‘good’ versus ‘fair’, ‘less well’ and ‘bad’. The continuous time variable ‘Trend (cont.)’ is coded 0 for 1994 and 1 for 2018. Reference group in model 1: 1994–1998, and in model 2: first year of observation (1994). Interaction term between family status (single and partnered mothers) and time (reference group = partnered mothers and first year of observation), 95% CI = 95% confidence interval, * *p* ≤ 0.05, ** *p* ≤ 0.01, *** *p* ≤ 0.001.

**Table 5 ijerph-19-02727-t005:** Development of socioeconomic factors in single and partnered mothers aged 30–49 years in Germany, 1994–2018.

Socioeconmic Factors	Time Trend			
	Single Mothers (*n_max_* = 13,664)		Partnered Mothers (*n_max_* = 70,179)	
	OR	95% CI	OR	95% CI
Education				
Low	0.51 **	0.32; 0.83	0.26 ***	0.21; 0.33
High	1.47	0.85; 2.56	2.86 ***	2.31; 3.55
Income				
Low	2.32 ***	1.69; 3.19	2.36 ***	1.89; 2.93
High	0.62	0.18; 2.12	1.21	0.97; 1.51
Occupational position				
Low	1.11	0.74; 1.66	0.76 **	0.63; 0.91
High	0.73	0.36; 1.47	2.00 ***	1.54; 2.60
Employment status				
Not employed	1.36	0.93; 1.98	0.57 ***	0.49; 0.67
Unemployed	1.85 **	1.25; 2.74	0.57 ***	0.44; 0.74
Part-time employed	1.50 *	1.02; 2.22	2.09 ***	1.81; 2.41
Full-time employed	0.48 ***	0.31; 0.74	0.70 ***	0.57; 0.85

Notes: Based on logistic regression analyses adjusted for age; OR = odds ratio; 95% CI = 95% confidence interval. For layout reasons, the continuous predictor (time trend 1994–2018) is listed in columns while the criterion variables (socioeconomic factors) are listed in rows. A regression model was calculated for each criterion variable. Reference category: first year of observation (1994), * *p* ≤ 0.05, ** *p* ≤ 0.01, *** *p* ≤ 0.001.

**Table 6 ijerph-19-02727-t006:** Development of socioeconomic factors (predicted probabilities) in single and partnered mothers aged 30–49 years, Germany, 1994–2018.

Socioeconomic Factors	1994–1998		1999–2003		2004–2008		2009–2013		2014–2018	
	%	95% CI	%	95% CI	%	95% CI	%	95% CI	%	95% CI
Single Mothers (*n_max_* = 13,664)										
Low education	38.0	0.29–0.47	32.5	27.4–37.7	32.2	26.6–37.9	23.7	20.8–26.6	27.4	23.2–31.4
High Education	17.0	10.4–23.5	19.1	14.4–23.7	20.1	15.1–25.2	24.4	21.2–27.6	21.5	18.2–24.9
Low income	25.6	20.5–30.7	30.2	26.4–34.1	39.6	35.2–44.0	38.6	35.9–41.3	41.6	38.1–45.1
High income	5.1	1.7–8.6	2.1	0.7–3.5	1.8	0.8–2.8	2.8	1.9–3.7	2.8	1.7–4.0
Low occupational position	14.3	20.0–18.7	19.2	15.1–23.3	19.5	15.8–23.1	17.1	14.9–19.4	16.8	14.1–19.5
High occupational position	12.6	7.13–18.1	13.2	9.3–17.1	7.3	5.0–9.6	11.1	9.3–12.9	11.0	8.9–13.1
Not employed	28.7	23.0–34.4	26.1	22.1–30.2	34.2	29.4–39.0	32.4	29.6–35.3	32.2	28.4–35.9
Unemployed	12.1	8.6–15.7	10.5	8.2–12.7	19.3	15.4–23.2	18.3	16.1–20.4	16.5	13.9–19.1
Part-time employment	31.9	24.7–39.1	35.3	30.8–39.8	38.4	33.7–43.1	39.8	36.9–42.6	39.2	35.4–3.0
Full-time employment	39.5	31.9–47.1	38.8	33.8–43.7	27.5	24.8–30.9	27.9	24.8–30.9	28.6	24.6–32.6
Partnered Mothers (*n_max_* = 70,179)										
Low education	35.5	32.7–38.3	27.1	25.2–29.0	23.8	21.4–26.1	18.1	16.2–19.9	15.6	13.8–17.4
High Education	18.0	15.6–20.3	22.1	20.3–23.9	26.5	24.2–28.9	29.9	27.9–32.0	35.1	32.8–37.3
Low income	5.4	4.6–6.2	5.8	5.0–6.5	7.1	5.9–8.2	8.1	7.3–8.9	10.3	9.3–11.4
High income	14.5	12.4–16.5	14.9	13.5–16.3	14.9	13.5–16.4	17.1	15.6–18.7	16.1	14.7–17.5
Low occupational position	14.3	12.8–15.8	16.9	15.5–18.3	16.3	14.6–18.0	13.6	12.2–15.0	12.0	10.9–13.2
High occupational position	9.3	7.7–11.0	9.5	8.4–10.7	10.0	8.6–11.3	12.6	11.2–14.0	15.0	13.5–16.5
Not employed	41.4	38.9–43.9	38.0	36.3–39.8	37.0	34.9–39.2	32.6	39.9–34.2	31.2	29.7–32.8
Unemployed	5.8	5.0–6.7	4.5	4.0–5.1	4.1	3.4–4.7	3.5	3.0–3.9	4.0	3.5–4.6
Part-time employment	35.7	33.3–38.1	42.9	41.1–44.6	47.1	45.0–49.3	50.4	48.6–52.3	50.2	48.3–52.1
Full-time employment	22.8	20.9–24.7	19.1	17.7–20.5	15.9	14.3–17.4	16.9	15.4–18.5	18.4	16.8–20.1

Notes: adjusted for age; % = predicted probabilities based on logistic regression analyses; 95% CI = 95% confidence interval. For layout reasons, the continuous predictor (time trend 1994–2018) is listed in columns while the criterion variables (socioeconomic factors) are listed in rows. A regression model was calculated for each criterion variable.

**Table 7 ijerph-19-02727-t007:** Decomposition of the total time effect on self-rated health (SRH) among single mothers aged 30–49 years into direct and indirect effects via socioeconomic factors, Germany 1994–2018.

**Decomposition**	**Good SRH**		**Poor SRH**	
	**OR**	**95% CI**	**OR**	**95% CI**
Total time effect	0.60 **	0.44; 0.83	2.06 **	1.37; 3.12
Direct time effect	0.70 *	0.51; 0.95	1.71 *	1.14, 2.56
Indirect time effect	0.87 *	0.78; 0.97	1.21 *	1.06; 1.38
	**APE (% points)**	**95% CI**	**APE (% points)**	**95% CI**
Total time effect	−12.0 **	−19.3; −4.7	9.9 ***	4.6; 15.3
Direct time effect	−8.6 *	−16.0; −1.2	7.3 **	2.0; 12.7
Indirect time effect	−3.4	_^1^	2.6	_^1^
**Conf_Pct**	28.2%		26.4%	
**Indirect effects**	**Coef**	**P_diff**	**Coef**	**P_diff**
Primary education	0.6	−16.4	−0.1	−2.0
Tertiary education	−0.2	4.7	0.1	2.3
Low occupat. Status	<−0.1	1.3	<0.1	0.7
High occupat. Status	−0.8	23.2	0.6	21.9
Income: <60%	−0.9	25.9	0.2	15.2
Income: >150%	−0.2	4.5	0.4	5.9
Full-time employed	−1.1	32.1	0.7	27.4
Unemployed	−0.8	24.8	0.7	28.6

Notes: Based on decomposition analysis using the KHB method, OR = odds ratio, reference group of time effect: first year of observation (1994), APEs = average partial effects (change in average predicted probability of good/poor SRH over time in percentage points), ^1^: 95% Confidence interval cannot be calculated since standard errors of indirect effects are not known for APE method, Conf_Pct = confounding percentage (proportion of the total effect that is due to all mediators) Coef: indirect effect of each of the mediators (in percentage-points), P_diff: contribution of each mediator to the indirect effect in percentages (the sum of all P_diff values adds up to 100 percent), * *p* < 0.05, ** *p* < 0.01, *** *p* < 0.001.

## Data Availability

The datasets used are available from the corresponding author. German data privacy laws necessitate that all users sign a data user contract with DIW Berlin.

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
