# Peer review of "Getting Better or Getting Worse? A Population-Based Study on Trends in Self-Rated Health among Single Mothers in Germany between 1994 and 2018"

_ijerph, 2022, doi:10.3390/ijerph19052727_

Round 1

Reviewer 1 Report

Dear Authors!

This valuable contribution fills a gap in the research on health. In spite of its limitations acknowledged by the authors themselves, I think these are not too plausible and they do not make the validity of the results questionable.

The results go against reasonable expectations and also against some evidence suggesting that the situation of single mothers has ameliorated recently due to the growing societal acceptance of this family form!

There are only two minor and one bigger issue I would like to raise:

  1. In line 97 it is stated that mothers were surveyed 13,664 and 70,179 times. However, in Table 1, the numbers are different: only slightly for single mothers, but the case number is 5,000 higher for partnered mothers. Could the authors address the reason of this change at least in a footnote, so that it becomes clear?
  2. It remains a little bit fuzzy for the reader, in Table 7, that the P_diff values, if added, result in 100%. Readers with statistical background must be knowledgeable of that, but I recommend including one sentence somewhere around lines 221-225.
  3. Most importantly, as in lines 257-258 it is stated, in accordance with the literature: ... "health disparities in single mothers did not completely disappear after controlling for socioeconomic factors". Although calculations are carried out with "hardcore" socioeconomic predictors, at least in one paragraph I suggest adding some aspects that go beyond them. I recommend two literatures, the first one on the subjective experiences of single parenthood, which suggest that, in spite of the growing acceptance and changing attitudes towards single parent families, being a single mothers can still be an emotionally heavy burden.

https://pubmed.ncbi.nlm.nih.gov/27150964/

Second, another predictor should be very briefly addressed: associated with socioeconomic features, health literacy at least partly mediates the relationship between socioeconomic features and health outcomes. Recent studies assess the health literacy disadvantage of single mothers controlled for socioeconomic disadvantages! 

https://www.mdpi.com/1660-4601/18/11/5517/htm

Please mention these two aspects, if possible around lines 257-260.

Congratulations for this valuable contribution!

Best wishes,

Reviewer

Author Response

First of all, we would like to thank the reviewers for the constructive comments and valuable suggestions on our manuscript. In revising our manuscript, we followed the comments as much as possible.

Reviewer 1

  1. In line 97 it is stated that mothers were surveyed 13,664 and 70,179 times. However, in Table 1, the numbers are different: only slightly for single mothers, but the case number is 5,000 higher for partnered mothers. Could the authors address the reason of this change at least in a footnote, so that it becomes clear?

Answer: Thank you for pointing out this discrepancy. It was a numerical error that has been corrected.

  1. It remains a little bit fuzzy for the reader, in Table 7, that the P_diff values, if added, result in 100%. Readers with statistical background must be knowledgeable of that, but I recommend including one sentence somewhere around lines 221-225.

Answer: The recommended sentences was included.

  1. Most importantly, as in lines 257-258 it is stated, in accordance with the literature: ... "health disparities in single mothers did not completely disappear after controlling for socioeconomic factors". Although calculations are carried out with "hardcore" socioeconomic predictors, at least in one paragraph I suggest adding some aspects that go beyond them. I recommend two literatures, the first one on the subjective experiences of single parenthood, which suggest that, in spite of the growing acceptance and changing attitudes towards single parent families, being a single mothers can still be an emotionally heavy burden.

Answer: The literature fits very well in this context, thank you! I have integrated them.

Reviewer 2 Report

This manuscript is a fascinating study on self-rated health in single mothers in Germany, similar to similar studies carried out in other European countries.

1. The analysis is correct, but grouping the temporal evolution in five-year periods may mask interesting information. Therefore, additional analyses should be performed. For the analysis of time trends, I suggest that a joint point regression With the global annual SHR data should be performed in addition to the analyses presented. To estimate global trends and detect if there have been changes in the trend at any time. Joint point regression is straightforward to do. It is only necessary to have the rates or percentage of SHR in each year. Joint Point regression can be done with the free program joinpoint https://surveillance.cancer.gov/joinpoint/. It is also possible to do it with STATA.

2. Please provide information on data availability. A panel reference is provided. You should indicate the data acquisition process. If the German Socioeconomic Panel Survey (G-SOEP) data are available on the web, you should give the link. Otherwise, you should provide information, e.g., GSOEP data can be used after writing to X, buying a CD with the data, etc.

3. Please explain in more detail what a rescaling bias consists of in general, and in this particular case, in a way that an unfamiliar reader will understand.

4. Indicate whether you used a macro or a command written by Stata users to perform the Karlson-Holm-Breen analysis. If so, indicate it in the material and methods and provide the bibliographic reference.

Author Response

First of all, we would like to thank the reviewers for the constructive comments and valuable suggestions on our manuscript. In revising our manuscript, we followed the comments as much as possible.

Reviewer 2

  1. The analysis is correct, but grouping the temporal evolution in five-year periods may mask interesting information. … Joint point regression is straightforward to do. It is only necessary to have the rates or percentage of SHR in each year

Answer: Thank you very much for this valuable comment! I found the program nlhockey in STATA and made first analyses with it. Due to time constraints, I unfortunately do not have enough time to deal sufficiently with this approach, which is new to me. I regret that! However, an analysis with smaller time intervals (3 years) has confirmed the results, thus they seem to be robust

  1. Please provide information on data availability. A panel reference is provided. You should indicate the data acquisition process. If the German Socioeconomic Panel Survey (G-SOEP) data are available on the web, you should give the link. Otherwise, you should provide information, e.g., GSOEP data can be used after writing to X, buying a CD with the data, etc.

Answer: After revision, this information is found in the manuscript.

  1. Please explain in more detail what a rescaling bias consists of in general, and in this particular case, in a way that an unfamiliar reader will understand.

Answer: The rescaling bias was explained in more detail after revision.

  1. Indicate whether you used a macro or a command written by Stata users to perform the Karlson-Holm-Breen analysis. If so, indicate it in the material and methods and provide the bibliographic reference.

Answer: After revision, this information is found in the manuscript.